# Effects of Heat Treatment on the Electromagnetic Wave Absorption Characteristics of Resorcinol Formaldehyde Silicon Dioxide Ceramic Particles

**DOI:** 10.3390/ma17102376

**Published:** 2024-05-15

**Authors:** Haiyang Zhang, Xinli Ye, Jianqing Xu, Shan Li, Xiaomin Ma, Wei Xu, Junxiong Zhang

**Affiliations:** 1Research & Development Institute of Northwestern Polytechnical University in Shenzhen, Shenzhen 518063, China; zhang15237660623@126.com (H.Z.); xjq15271972321@163.com (J.X.); li_shan@nwpu.edu.cn (S.L.); 2School of Civil Aviation, Northwestern Polytechnical University, Xi’an 710072, China; 3Yangtze River Delta Research Institute of Northwestern Polytechnical University, Suzhou 215400, China; 4National Equipment New Material & Technology (Jiangsu) Co., Ltd., Suzhou 215400, China; 13914089968@163.com; 5Xi’an Electronic Engineering Research Institute, Xi’an 710100, China; 6School of Textile and Clothing, Nantong University, Nantong 226019, China; zhangjunxiong126@163.com

**Keywords:** electromagnetic wave absorbing, SiO_2_ ceramic particles, sol–gel method

## Abstract

In light of the pressing environmental and health issues stemming from electromagnetic pollution, advanced electromagnetic wave absorbing materials are urgently sought to solve these problems. The present study delved into the fabrication of the resorcinol formaldehyde (RF)/SiO_2_ ceramic particles using the sol–gel route. From SEM images and XRD and XPS analysis, it can be seen that the RF/SiO_2_ ceramic particles are successfully generated after heat treatment at 1500 °C. At room temperature, the sample treated at 1500 °C exhibited a minimum reflection loss of −47.6 dB in the range of 2–18 GHz when the matching thickness was 5.5 mm, showcasing strong attenuation capabilities. Moreover, these particles demonstrated a considerable effective electromagnetic wave absorption bandwidth of 3.14 GHz, evidencing their potential for wideband electromagnetic wave absorption. The temperature adjustment played a pivotal role in achieving optimal impedance matching. When the heat treatment temperature is increased from 800 °C to 1500 °C, the dielectric properties of the material are improved, thus achieving the best impedance matching, thereby optimizing the material’s absorption properties for specific frequency ranges, which makes it possible to customize the electromagnetic wave-absorbing characteristics to meet specific requirements across a range of applications.

## 1. Introduction

Given the rising prevalence of electronic devices and wireless communication systems, electromagnetic interference (EMI) is now a significant concern [1,2,3,4,5], ranking alongside traditional forms of pollution like water and air pollution. EMI can disrupt the operation of electrical equipment and has potential implications for human health. To mitigate these effects, absorbent materials are being researched and developed [6,7,8,9,10]. These materials must exhibit specific characteristics to effectively address EMI. Firstly, they should be lightweight to facilitate practical applications across various contexts [11,12,13,14,15]. Additionally, excellent absorption properties are crucial to efficiently attenuate electromagnetic waves. A wide effective bandwidth ensures the ability to mitigate a broad range of frequencies, enhancing their versatility and effectiveness [16,17,18]. Moreover, stability in performance over time is essential for consistent and reliable EMI mitigation. Meeting these criteria, absorbent materials can significantly minimize the impact of electromagnetic interference on technological systems and human well-being, especially in the military field such as radar cross-sections [19,20,21].

Silicon dioxide (SiO_2_) ceramic particles are recognized for their transparency and low complex dielectric constants, attributes that render them well suited for fine-tuning impedance matching in electromagnetic applications. Building upon this foundation, Jia et al. developed a dual-core–shell structured composite absorbent denoted as Fe_3_O_4_@SiO_2_@C/Ni, which possessed a minimum reflection loss (*RL*_min_) of −38.9 dB with a maximum effective bandwidth (*EAB*_max_) of 10.1 GHz. Through strategic adjustments to the thickness of both the carbon and SiO_2_ layers within the composite structure, the researchers could precisely modulate the absorption performance, which facilitated the attainment of optimal impedance matching, thereby amplifying the material’s effectiveness in attenuating electromagnetic waves (*EMWs*) across a wide spectrum of frequencies [22]. Zheng et al. employed the chemical vapor infiltration (CVI) method to infiltrate nano-crystalline SiC coatings into porous Si_3_N_4_ ceramics, resulting in the fabrication of porous Si_3_N_4_-SiC composite ceramics. Subsequently, they oxidized these Si_3_N_4_-SiC composite ceramics in the air at 1100 ℃ to generate a SiO_2_ film layer on the surface of the rod-like Si_3_N_4_-SiC grains, producing a Si_3_N_4_-SiC/SiO_2_ composite, which exhibited a multilayered structure, effectively reducing impedance mismatching and showcasing excellent electromagnetic absorption properties. Specifically, the reflection loss (*RL*) of the Si_3_N_4_-SiC/SiO_2_ composite material was measured at −30 dB at a frequency of 8.7 GHz and a thickness of 3.8 mm [23]. In their groundbreaking study, Wang et al. innovatively incorporated SiO_2_ fillers into photo-sensitive pre-ceramics, marking the first instance of reducing pyrolysis shrinkage using this approach. By adjusting the impedance-matching properties of SiOC ceramics, they effectively enhanced the electromagnetic absorption performance, which featured an *RL_min_* of −28.1 dB at a thickness of 1.9 mm and a notable electromagnetic wave absorption bandwidth (*EAB*) of 6.38 GHz at a thickness of 2.3 mm [24]. This study underscored the significance of engineering techniques in optimizing electromagnetic wave absorption properties, which laid the groundwork for the development of advanced materials with enhanced electromagnetic absorption performance, offering significant potential for various applications.

Current research on SiO_2_ ceramic particles is primarily concentrated on their ability to modulate material dielectric properties. Additionally, attention is directed towards addressing limitations associated with their application in high-temperature electromagnetic absorption fields, particularly concerning potential sintering phenomena at elevated temperatures. Modifying SiO_2_ ceramic particles has shown promise in enhancing their electromagnetic absorption performance. By refining their composition or structure through various techniques, researchers aimed to optimize their electromagnetic absorption capabilities while mitigating issues related to high-temperature sintering. Ye et al. synthesized C/SiC network-reinforced SiO_2_ aerogel composites via chemical vapor deposition, sol–gel, and high-temperature pyrolysis processes. The prepared aerogel exhibited a compressive stress of 3.46 MPa at a strain rate of 9.49%. Additionally, the material demonstrated an *EAB_max_* of 8.16 GHz at a matching thickness of 4.1 mm [25]. Xu et al. proposed a new strategy of constructing magnetic nickel nanochannels in SiC/SiO_2_ composite foams for the synthesis of Ni/SiC/SiO_2_ porous composites (NSPCs). The NSPC can obtain an extremely strong reflection loss of 64.86 dB (1.46 mm), the EAB covers 13.2 GHz (4.8–18 GHz), and the total thickness is 1.79 mm (1.21–3 mm), accounting for 82.5% of the entire frequency range [21]. Dong et al. prepared novel SiC-nanowire-reinforced SiO_2_/3Al_2_O_3_·2SiO_2_ porous ceramics by the precursor osmotic pyrolysis (PIP) method. SiC-nanowire-reinforced SiO_2_/ 3Al_2_O_3_·2SiO_2_ composite ceramics exhibit excellent electromagnetic wave absorption (PIP5) when the content of SiC nanowires is 23.9%. At 10 GHz, the composite ceramics have an *RL*_min_ of −30 dB, corresponding to more than 99.9% of the electromagnetic wave consumption. At a thickness of 5 mm, the EAB covers the frequency range of 8.2–12.4 GHz (the entire X-band) [26].

In contrast to pure SiO_2_ ceramic particles, composites or modifications have shown the potential to effectively enhance both electromagnetic absorption performance and high-temperature stability. However, challenges persist, including complex preparation processes and high-cost equipment requirements. To address these challenges, this study focused on modifying SiO_2_ ceramic particles using resorcinol formaldehyde (RF). Sol–gel technology was employed to introduce RF and construct a porous framework within the SiO_2_ particles. The RF/SiO_2_ ceramic particles were then fabricated using atmospheric pressure drying followed by high-temperature treatment at 800 and 1500 °C, which was simple and low-cast. Microstructural analysis and characterization were further conducted by SEM, XRD, and XPS, and we evaluated the electromagnetic wave absorption (*EMA*) performance, supporting the multi-function and multi-scenario application of RF/SiO_2_ ceramic particles. This approach aims to overcome the limitations associated with traditional SiO_2_ ceramic particles and pave the way for advanced materials with improved high-temperature resistance and electromagnetic absorption properties. In this study, the overall preparation process employed is both simple and low-cost. RF/SiO_2_ ceramic particles, prepared using readily available materials, were dried at atmospheric pressure, thus effectively avoiding cumbersome drying processes such as freeze-drying or supercritical drying.

## 2. Experimental Section

### 2.1. Materials

Anhydrous ethanol (AR, EtOH), isopropanol (AR), aluminum isopropoxide (AR), and resorcinol (AR, solid, R) were produced by Ron Reagent Limited Company (Shanghai, China). 3-aminopropyl triethoxysilane (AR, APTES) and tetraethyl orthosilicate (AR, TEOS) were supplied by Shanghai Aladdin Company. Formaldehyde (37–40% aqueous solution, F) was manufactured by Da Mao Chemical Reagent Factory (Shanghai, China).

### 2.2. Preparation of RF/SiO_2_ Ceramic Particles

The R, F, isopropanol, APTES, TEOS, and deionized water were mixed in a molar ratio of 1:2:23:1.8:1.2:4, with a corresponding n(C):n(Si) ratio of 1:3. The mixture, containing 3 g of aluminum isopropoxide, was stirred at 80 °C for 30 min and then sealed in the beaker, reacting at 60 °C for 24 h. Organic wet ceramic particles were obtained after aging at 60 °C for 48 h. These wet ceramic particles were then immersed in isopropanol for substitution at 60 °C for 48 h, with the isopropanol changed every 24 h. The solvent in the substituted wet gels was removed at room temperature and atmospheric pressure until no further mass change was observed, resulting in the dried RF/SiO_2_ ceramic particles, designated as RS-0. RS-0 was pyrolyzed in a N_2_ atmosphere with a flow rate of 100 mL/min, heated at a rate of 4 °C/min to 800 °C and held for 2 h, producing RS-800. Subsequently, RS-1500 was obtained by heating RS-0 at a rate of 4 °C/min to 1500 °C and held for 2 h, meaning that the final sample was stable and temperature-resistant [27].

### 2.3. Characterization

The morphology and structure of the aerogel samples were examined using a Hitachi SU8010 field-emission scanning electron microscope (SEM, Hitachi, Japan). Phase compositions were analyzed using a PANalytical X’Pert Pro X-ray diffractometer (XRD, Buker, Seattle, WA, USA). Chemical composition and atomic valence state analysis was conducted using the Thermo ESCALAB 250xi X-ray photoelectron spectrometer (XPS, ESCALAB, Waltham, MA, USA). Electromagnetic parameters were measured using an Agilent 5324A vector network analyzer (VNA, 5324A, Santa Clara, CA, USA) in the range of 2–18 GHz. The coaxial method was used to test the electromagnetic wave absorption performance, wherein the samples were mixed with paraffin wax at a mass ratio of 3:7 and formed into concentric rings with an outer diameter of 7 mm and an inner diameter of 3 mm. Finally, the minimum reflection loss was calculated via the transmission line theory. The *EMA* capacity of RF/SiO_2_ ceramic particles was evaluated via RL. RL could be calculated using Equations (1) and (2) [28]:(1)$$RL=20log\left|\frac{z_{in}-z_{0}}{z_{in}+z_{0}}\right|$$
(2)$$Z_{in}=Z_{0}{\left(\frac{\mu_{r}}{\varepsilon_{r}}\right)}^{0.5}tanh\left(j\left(\frac{2\pi fd}{c}\right){\left(\mu_{r}\varepsilon_{r}\right)}^{0.5}\right)$$
where *Z_in_* represents the input impedance of the absorber in air; *Z*_0_ is the free space impedance; *c* is the speed of light; *d* represents the matching thickness of the absorber; and *f* is the electromagnetic frequency. *ε_r_* and *μ_r_* are the dielectric constant and relative complex magnetic permeability of the absorber, respectively.

## 3. Results and Discussion

### 3.1. Structural Control of RF/SiO_2_ Ceramic Particles

In the context of the carbothermal reduction process, it was crucial to note that the carbothermal reduction temperature had a substantial impact on both the microstructural development and physical attributes of the RF/SiO_2_ ceramic particles. Consequently, to delve into these underlying mechanisms, a meticulous examination was carried out on the morphological and structural changes occurring at varying carbothermal reduction temperatures. Initially, when at ambient temperature, non-crystalline SiO_2_ was not formed in RS-0. The gel exhibited a uniform continuous portion internally, with localized particle aggregation and relatively weak inter-particle bonding, as depicted in Figure 1a. As the carbothermal reduction temperature increased to 800 ℃, however, these particles started aggregating and agglomerating, a phenomenon visible in Figure 1b. Further escalation of the carbothermal reduction temperature to 1500 ℃ led to the surfaces of the RF/SiO_2_ appearing notably smoother. This transformation could be attributed to the enhanced crystallinity of the RF/SiO_2_ ceramic particles achieved at such an elevated temperature, as illustrated in Figure 1c [29].

To delve into the phase transition dynamics, a crystal phase analysis was performed on samples RS-0, RS-800, and RS-1500, with their respective XRD patterns displayed in Figure 1d. In the case of RS-0, a broad diffraction peak within the range of 15 to 25° signified the presence of the amorphous SiO_2_ ceramic particles. The absence of any sharp, unique peaks in the XRD pattern of RS-800 following an increase in heat treatment temperature to 800 ℃ suggested that the ceramic particles retained their amorphous structure at this temperature level. However, upon subjecting the sample to a higher temperature of 1500 ℃, a prominent sharp peak emerged at 21.9°, which corresponded to the (101) crystal plane of SiO_2_. This observation conclusively indicated that a phase transformation had indeed occurred following the 1500 ℃ heat treatment [24]. Coupling this with the microscopic structural analysis, it was revealed that the initially amorphous SiO_2_ particles underwent crystallization at the relatively high temperature of 1500 ℃, forming a crystalline layer covering the surface of the material’s inherent framework.

The FTIR spectra of the RF/SiO_2_ ceramic particles are shown in Figure 1e. The peaks at 2887 cm^−1^, 1608 cm^−1^, and 1463 cm^−1^ were assigned to C-H stretching vibrations [30]. The peaks at 1062 cm^−1^ were attributed to the Si-O stretching and bending vibrations from SiO_2_ [31,32]. The peak height of the Si-O bond increased with the heat treatment temperature rising, which proved the crystallinity enhancement. Additionally, XPS analysis was employed to investigate the chemical structure of the products obtained at different heat treatment temperatures, particularly quantitatively analyzing the structural changes of silicon elements at different heat treatment temperatures. As shown in Figure 1f–h, all three samples exhibited distinct O-Si-O peaks, indicating the formation of SiO_2_ [28,33]. The peak intensity of the O-Si-O bonds increases with the rise in heat treatment temperature, while the corresponding peak intensity of the C-Si-O bonds begins to decrease, suggesting the conversion of more C-Si-O bonds into O-Si-O bonds, thus demonstrating enhanced crystallinity.

### 3.2. EMA Performance of RF/SiO_2_ Ceramic Particles

When *RL* reached −10 dB, it corresponded to *EMA* efficiency of 90%, and materials with *RL* values less than −10 dB fell into the range of *EAB*, meaning they absorbed electromagnetic waves efficiently [34]. Figure 2 illustrates the correlation between the *RL* values and changes in thickness and frequency for RS-0, RS-800, and RS-1500. The thicknesses under investigation spanned from 1 to 7 mm, and the frequencies tested ranged from 2 to 18 GHz. In Figure 2a, it was evident that RS-0 exhibited poor *EMA* performance, which might stem from its less favorable dielectric characteristics. However, when subjected to heat treatment, these samples demonstrated significantly better *EMA* performance at higher frequencies. Figure 2b,e demonstrates that RS-800 achieved an EAB of 6.01 GHz and an *RL*_min_ value of −22.54 dB. The arrows in the figures indicate the position of the *RL*_min_ value, indicating excellent *EMA* capability under the test conditions. Specifically, within the thickness range of 1.6 to 7 mm and the frequency range of 3 to 18 GHz, the material showed *RL* values consistently below −10 dB, indicating effective absorption over this spectrum. Turning to Figure 2c, it was observed that RS-1500 displayed an exceptional *EMA* property across a wide frequency band. Moreover, RS-1500 achieved a significantly broader *EAB* of 3.14 GHz and an *RL*_min_ value of −47.6 dB, reflecting its enhanced *EMA* capacity.

To graphically represent the enhancement in the *EMA* capabilities following heat treatment, Figure 3 showcases the *RL* curves of RS-1500 at various matching thickness levels. As shown in Figure 3a, when subjected to a heat treatment at 1500 °C, RS-1500 achieved its optimal performance with a peak *RL* of −47.6 dB, occurring at a thickness of 5.5 mm and a resonance frequency of 17.04 GHz. Additionally, the *RL* values maintained a level below −10 dB across two frequency bands—4.56 to 5.52 GHz and 14.58 to 16.76 GHz—when the thickness was set at 6 mm. This resulted in an *EAB* value of 3.14 GHz, illustrating a broadened absorption capacity. As evidenced in Figure 3b, the peak *RL* of RS-1500 consistently decreased as the matching thickness increased, aligning with the principles of the quarter-wavelength theory, which suggested that the maximum absorption occurred when the material’s thickness corresponded to a quarter of the incident electromagnetic wave’s wavelength [28]. The dashed line in Figure 3b represents the minimum reflection loss at different matching thicknesses, corresponding to points on the quarter-wavelength line one by one, in accordance with the quarter-wavelength theory. This finding highlights the fact that the *EMA* properties of RS-1500 could be effectively manipulated and optimized through heat treatment processes, enabling it to maintain strong absorption performance across a range of thickness dimensions. By tuning the thickness, RS-1500 could exhibit a broad bandwidth of effective absorption within specific frequency ranges. Thus, RS-1500 demonstrated significant potential as a versatile and tunable material for applications requiring controlled or enhanced electromagnetic wave attenuation.

Optimization of the impedance matching (*|Z_in_/Z_0_|*) results in excellent EMW absorption performance. The *|Z_in_/Z_0_|* value of one indicates that the absorber has a great impedance match and lets EMWs easily enter inside. The attenuation constant (*α*), as another influencing factor, indicates that the greater its value, the stronger the absorber’s ability to attenuate *EMWs* [35]. In Figure 4, an expanded representation is provided that elucidates the intricate relationship between *RL*, *|Z_in_/Z_0_|*, and *α* when the thickness of RS-1500 is specifically adjusted to 5.5 mm. Upon attaining this matching thickness and operating at a frequency of 17.04 GHz, the ratio of the input impedance to the characteristic impedance of free space for RS-1500 nearly reached the value of one. This close approximation to perfect impedance matching was highly significant because it indicated that most of the incoming electromagnetic waves encountered minimal reflection at the interface of RS-1500 and were efficiently coupled into the material’s interior. This optimal impedance match allowed for maximal transmission and subsequent absorption of the electromagnetic energy. Although impedance matching reaches one around frequencies of 5 and 7 GHz, the corresponding attenuation constants are relatively small, resulting in failure to achieve lower reflection loss values. In conjunction with the SEM images of RS-1500, the crystallization phenomenon of the initial amorphous SiO_2_ particles, resulting in the formation of a crystalline layer covering the surface of the material’s inherent framework, significantly enhanced the excellent impedance matching and attenuation constants. Therefore, with this unique combination of thickness and frequency, RS-1500 demonstrated an exceptional *EMA* performance, where it facilitated a profound reduction in the intensity of electromagnetic waves passing through it. This phenomenon not only substantiated the material’s superior *EMA* capabilities following meticulous thickness optimization but also served as a critical reference point for the design and application of advanced microwave-absorbing materials in targeted frequency regimes.

Figure 5a,b illustrate the impact of rising heat treatment temperatures on the real (ε′) and imaginary (ε″) part of the complex permittivity, respectively, which was directly associated with the ability to store electromagnetic energy. The ε′ of RS-800 was greater than RS-0 and RS-1500, indicating that the energy storage capacity of the material first increases and then decreases with the increase in heat treatment temperature. The fluctuations observed in the ε′ curve within the 13–15 GHz range could be attributed to the alignment of certain randomly oriented dipoles in the ceramic particles along the direction of the electromagnetic field, leading to dipole polarization effects. Over the entire frequency spectrum of 2–18 GHz, the complex permittivity (ε″) of RS-800 was also greater than that of RS-0 and RS-1500. This indicates that the energy dissipation inside the material first increases and then decreases with the increase in temperature. Due to the inherent dispersion property of the material, both ε′ and ε″ showed a decreasing trend with increasing frequency when subjected to a heat treatment temperature of 800 ℃. In the high-frequency domain from 15 to 18 GHz, ε′ remained relatively steady despite changes in frequency for all materials treated at different temperatures. The dielectric loss tangent (tan*δ*_E_ = ε″/ε′) serves as a measure of a material’s energy dissipation efficiency. As portrayed in Figure 5c, the trend in tan*δ*_E_ mirrors that of ε″ across the 6–18 GHz range. Notably, within the lower frequency interval of 2–6 GHz, the tan*δ*_E_ of RS-1500 rose with increasing frequency. However, a contrasting behavior was observed in the higher frequency span of 14–17 GHz, where tan*δ*_E_ decreased as the frequency increased, suggesting that the rate of energy dissipation became less pronounced at 14–17 GHz. The EMA performance of the material is not only determined by the dielectric constant, but also by its impedance matching. According to the previous impedance matching analysis, although the dielectric constant of RS-800 is larger than that of RS-1500, the impedance matching of RS-1500 is closer to one, demonstrating better EMA performance.

The above results show that the RF/SiO_2_ ceramic particles treated at 1500 °C show abnormal equilibrium performance, and the *RL*_min_ reaches −47.6 dB, which indicates that it has an excellent electromagnetic wave attenuation ability. In addition, at a thickness of 6 mm, these particles exhibit a basic *EAB* of 3.14 GHz. This study not only contributes to the production of a new class of high-performance *EMA* materials, but also provides valuable guidance for the further optimization and design of advanced *EMA* materials.

## 4. Conclusions

(1)In conclusion, this study focused on modifying SiO_2_ ceramic particles using RF. Sol–gel technology was employed to introduce RF and construct a porous framework within the SiO_2_ particles. The RF/SiO_2_ ceramic particles were then fabricated using atmospheric pressure drying followed by high-temperature heat treatment. Microstructural analysis and characterization were further conducted.(2)A systematic examination was conducted to explore the impact of heat treatment temperatures on *EMA* performance. The results showed that the RF/SiO_2_ ceramic particles treated at a temperature of 1500 °C displayed an exceptionally balanced performance, achieving an *RL*_min_ of −47.6 dB, which was indicative of superior electromagnetic wave attenuation capability. Furthermore, at a thickness of 6 mm, these particles presented a substantial *EAB* of 3.14 GHz.(3)This study not only contributed to the production of a new class of high-performance *EMA* materials but also offered valuable guidelines for further optimizing and designing advanced *EMA* materials.(4)The findings emphasized the potential of the sol–gel technique combined with precise thermal treatments in fabricating innovative, high-efficiency *EMA* solutions for contemporary and emerging technological needs.

## Figures and Tables

**Figure 1 materials-17-02376-f001:**
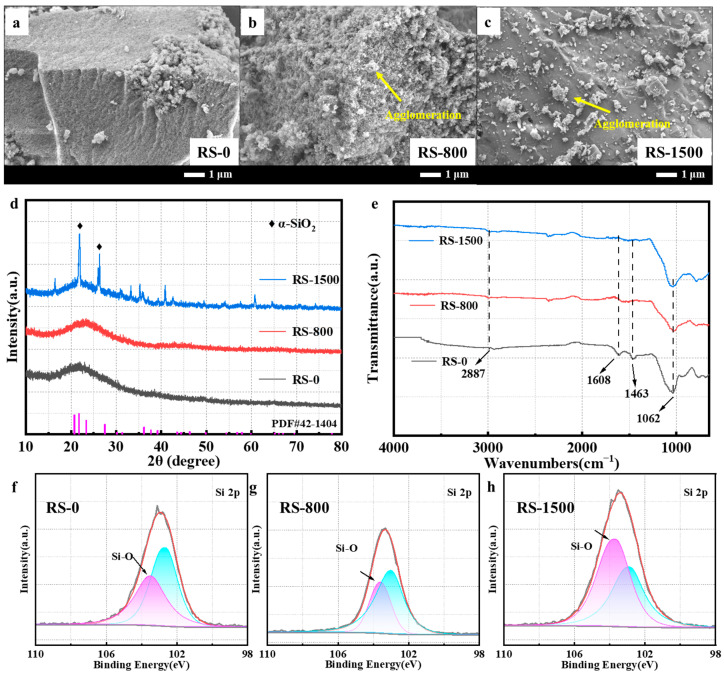
SEM for RS-0 (**a**), RS-800 (**b**), and RS-1500 (**c**); XRD of all samples (**d**); FTIR of all samples (**e**); XPS for RS-0 (**f**), RS-800 (**g**), and RS-1500 (**h**).

**Figure 2 materials-17-02376-f002:**
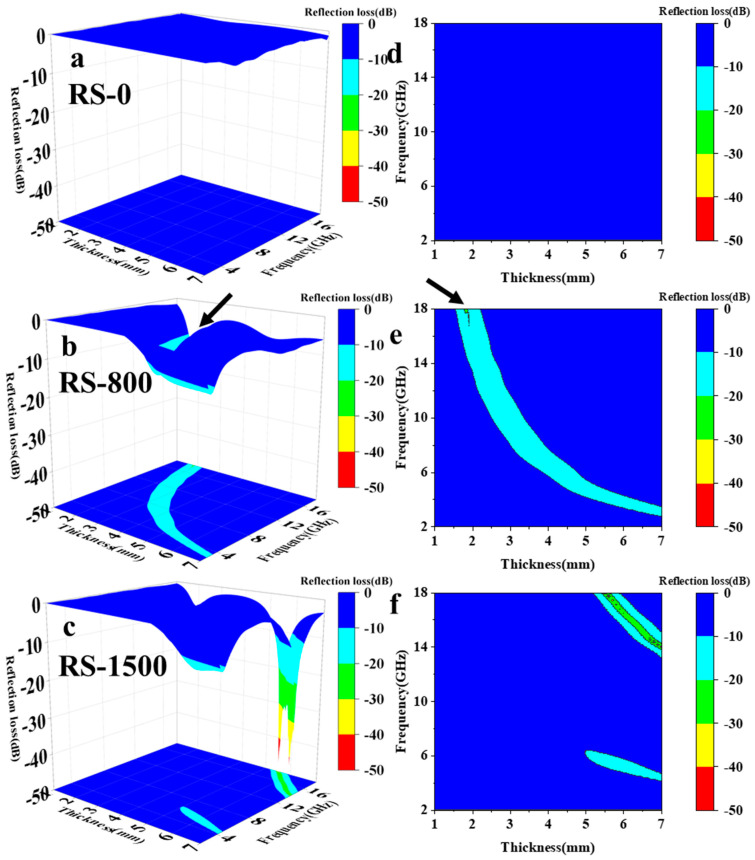
*EMA* performance of RS-0 (**a**,**d**), RS-800 (**b**,**e**), and RS-1500 (**c**,**f**).

**Figure 3 materials-17-02376-f003:**
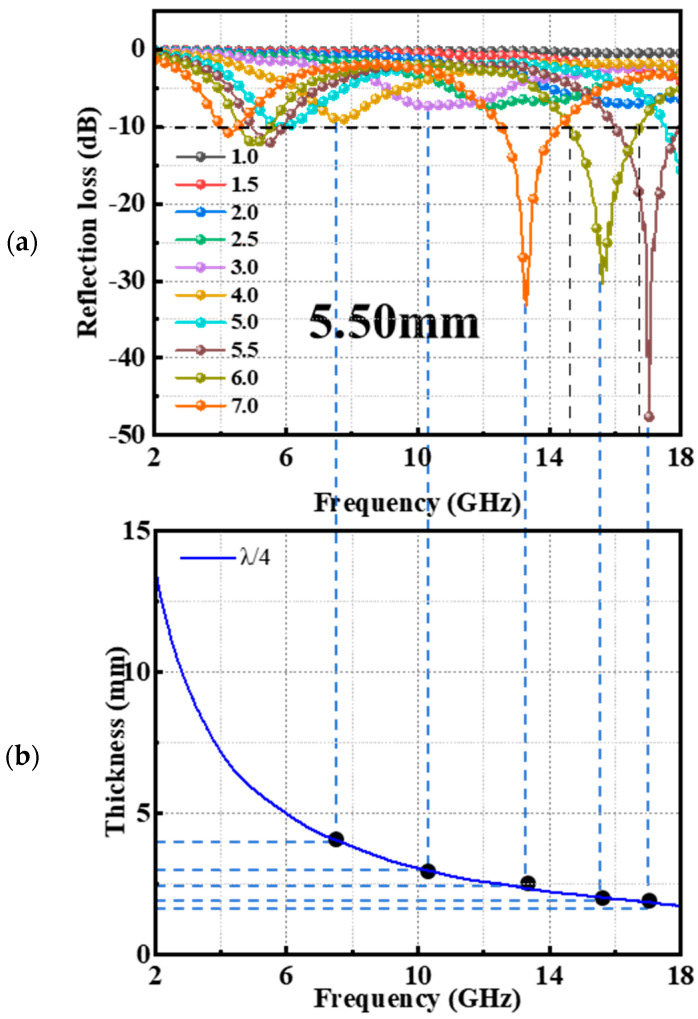
*EMA* performance of RS-1500: *RL* curve at different matching thicknesses (**a**); the relationship between match thickness and peak frequency (**b**).

**Figure 4 materials-17-02376-f004:**
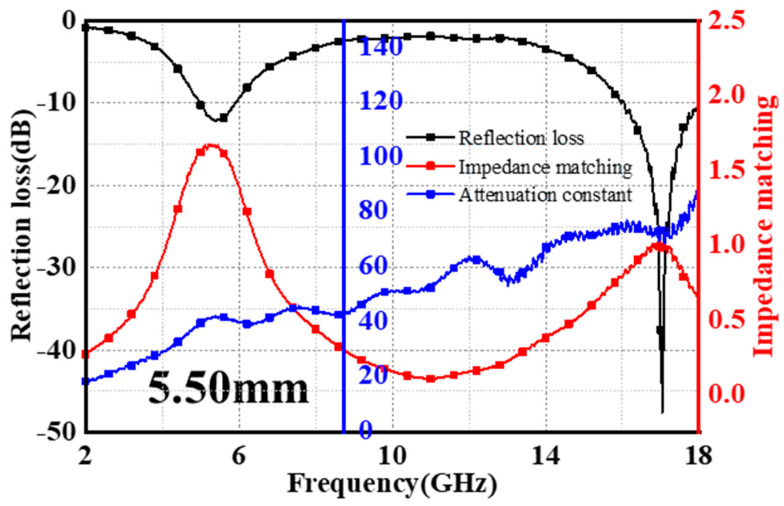
Relationship between *RL*, impedance matching, and attenuation constant.

**Figure 5 materials-17-02376-f005:**
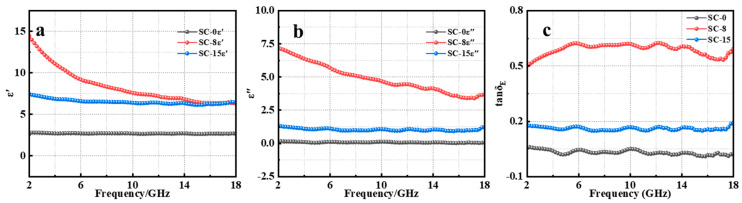
Electromagnetic parameters of all samples: real part of complex permittivity (**a**); imaginary part of complex permittivity (**b**); dielectric loss tangents (**c**).

## Data Availability

The data can be provided if requested.

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
