# Peer review of "Effects of Heat Treatment on the Electromagnetic Wave Absorption Characteristics of Resorcinol Formaldehyde Silicon Dioxide Ceramic Particles"

_materials, 2024, doi:10.3390/ma17102376_

Round 1
Reviewer 1 Report
Comments and Suggestions for Authors
The manuscript was submitted without lines number identification which makes it very difficult for the review. This should have been addressed prior to review.
18 references are insufficient, typical manuscripts in this journal for latest articles reference over 30 articles. See DOIs (10.3390/ma17081858, 58 references), (10.3390/ma17081857, 56 references), and (10.3390/ma17081855, 58 references).
Why do the references have [J] after each title?
A better title might be “Effects of heat treatment on the electromagnetic wave absorption characteristics of resorcinol formaldehyde silicon dioxide ceramic particles”
The authors need to provide a hypothesis as to what factors were responsible for the observed excellent absorption properties of the RF/SiO2 particles.
There is no discussion of stability of the EM wave absorbers? This is a major issue in the field.
There is no discussion of structural defect sites within absorbers as defects sites are often key contributor to EM wave loss.
Abstract
“was becoming” is not correct grammar
Define RF prior to use
Describe the specific aspect of the fabrication that resulted in a “record-low minimum reflection loss of -47.60 dB” was it particle treatment at a particular temperature or particle composition?
Describe the temperature adjustment (e.g. change from 800 to 1500 degrees C)
State if these exciting properties (strong attenuation, impedance matching) only occur at specific temperatures or remain after temperature application is removed
State the specific frequency ranges (i.e. 2-18 GHz)
Include the methods used to characterize the particles (scanning electron microscopy, XRD, XPS, etc.) and what was identified
Introduction
Page 1
Provide specific examples of environmental and health issues than stem from electromagnetic pollution. Just saying “has potential implications for human health” is too broad, vague, and not definitive.
“absorbent are being researched and developed” does not make sense, do the authors mean absorbent materials, if so absorbent to what frequencies/wavelengths and cite examples what is being developed and why.
“absorbent can” does not make sense, do the authors mean absorbent materials?
Define as silicon dioxide (SiO2)
Page 2
“Subsequently, they” is a run-on sentence
Define RL as reflection loss for the reader prior to use
After citing reference 12 a sentence is needed as to provide the reader with the importance of the referenced work and how this relates to the presented study (before introducing another study).
Define EAB as effective absorption bandwidth
After “Modifying SiO2 ceramic particles has shown promise in enhancing their electromagnetic absorption performance.” More examples besides the Ye et al. paper are needed as to what modifications have resulted in performance alterations. Clarify if this example is a ceramic aerogel due to the compressive stress.
When the authors say “and high-temperature resistance” what precisely do they mean, resistance to oxidation? corrosion? scaling? All?
When the authors say “high equipment requirements” what precisely do they mean, high cost? power? size?
The authors need to address the human health risk factors associated with resorcinol formaldehyde, is this a mixture of two liquids or a polymer? aerogel?
“high-temperature heat treatment” is redundant, better to say “high-temperature treatment at 800 and 1500 degrees C”
State the types of microstructural analysis and characterizations performed
Experimental Section
The A in aluminum isopropoxide should be lower case
from where was the paraffin wax obtained?
Page 3
APTE should be APTES
How was the mixture sealed in the beaker? Using parafilm?
“when subjected to lower temperatures” What do the authors mean? It was the readers understanding that RS-0 was RT, should this say “ambient temperatures” or do the authors mean “temperatures below 800” ?
How are the particles “bonding together” do the authors mean coalescence? Agglomeration? Dimerization? How is this phenomenon visible in B? Add arrows to identify.
SEM is not the technique to identify quantity: how did increasing the temp to 1500 ℃ yield a greater quantity of the RF/SiO2 ceramic particles? Were the particles smaller and/or the mass yield more?
Optical photos of the RS-0, RS-800, and RS-1500 along with the SEM images would be helpful.
Include the substrate used for XRD. If glass, could the broad 15-25 degree peak be due to background from the substrate?
Add discussion regarding the presence of SiC, include the standard diffraction data for alpha-SiO2 and beta-SiC and the JCPDS card number
Page 4
“was shown” should be “is shown”
There is no discussion regarding differences in the FTIR spectra for the three different samples and what the spectra indicate with regards to material composition/properties. If no differences why present this data?
Identify that the spectra in Fig 1E are off-set for clarity
Identify the blue spectra in Fig 1 f-h and discuss peak width
Bottom of page 4
Define EMA as electromagnetic absorption prior to use
“an RL” is not correct
Should be Figure 2 illustrates
Page 5
Explain what you mean by “at higher frequencies” frequency >3 GHz?
Shouldn’t an RLmin value of -22.54 dB be shown as green not teal in Fig 2 b,e?
Figure 2 Thickness and frequency are too difficult to read in a-c, also it loos like the axis starts at 1.5 mm not 1.0 mm as described in the text
Page 6
Should it be written as “resulted in EAB values of 3.14 GHz and 15.7 GHz, respectively”?
Discussion should be split and fig 3a and b referenced separately in the text, perhaps change to “As evidenced by Figure 3b, the peak”
Fig 3. Caption, do the authors mean “curves at” different thicknesses instead of “curve under”? what do the authors mean by “Match the relationship”?
Fig 3. What do the dashed vertical lines represent?
Should be “an expanded representation is provided that elucidates”
Page 7
Isn’t the value of 1.0 impendence matching also reached at frequencies of 5 and 7 GHz, why is this not discussed as well? So, it should instead be written “this unique combination of thickness and frequencies”?
Should be “Figures 5a and 5b illustrate the impact of rising heat treatment temperatures on the real (ε') and imaginary (ε″) part of the complex permittivity, respectively, which was directly associated with the ability to store electromagnetic energy”
Comment on why RS-800 has higher permittivity than RS-1500. Are the labels incorrect in Fig 5 (blue and red mislabeled)? The “the imaginary part of the complex permittivity (ε'')” does not “increase with escalating heat treatment temperatures” the discussion does not match the figures.
Page 8
The research didn’t successfully synthesize, the research demonstrates successful synthesis of
Comments on the Quality of English Language
There are numerous tense issues throughout the manuscript as identified above.
Anonyms are not introduced properly
Reviewer 2 Report
Comments and Suggestions for Authors
Dear authors,
the manuscript is very interesting however, I have some comments and suggestions which could improve the manuscripts quality.
1. In the introduction section please emphasize the novelty. It should be in a separate paragraph. Also include hypothesis of your research in explicit format - preferably in bullet format. The last paragraph of the introduction section should contain a outline of the manuscript i.e. short description of each section in the manuscript.
2. Description of materials and methods should be expanded as much as possible. Since this manuscript delved into the fabrication process of the RF/SiO2 the process should be described (if possible). Also, the materials and methods section should contain a description of the measurement process of minimum reflection loss in detail.
3. The results and discussion sections should be separated. In the result section you should present results only. The discussion section should contain a detailed analysis of the obtained results.
4. All the graphs for example in Figure 1 should contain girds to improve readability. Also, the y-axis should contain numbers of intensity variables. All graphs in the manuscripts should contain grids.
5. The conclusion is too short and weak in terms of structure. The first paragraph should contain a short description of what was done in the manuscript. The second paragraph (in bullet format) should contain the answers to hypotheses defined in the introduction section that are based on the detailed discussion section. The third paragraph should contain the advantages and disadvantages of the proposed research methodology based on which the fourth paragraph i.e. future work should be written.
Reviewer 3 Report
Comments and Suggestions for Authors
1) Since it is an important field, it is necessary to expand the base of works studied, as well as the references used.
2) It is not good to pile up the citations [1-10], in a respected work they are discussed separately or grouped in twos and threes.
3) At the end of the introduction, which must be supplemented with more cited studies, briefly introduce the presentation of the following sections of the work.
4) The statements in section 3.1 require citations: "In the context of the carbothermal reduction process, it was crucial to note that the
carbothermal reduction temperature had a substantial impact on both the microstructural development and physical attributes of the RF/SiO2 ceramic particles. Consequently, to delve into these underlying mechanisms, a meticulous examination was carried out on the
morphological and structural changes occurring at varying carbothermal reduction temperatures. Initially, when subjected to lower temperatures, non-crystalline SiO2 was not formed in RS-0. The gel exhibited a uniform continuous portion internally, with localized
particle aggregation and relatively weak inter-particle bonding, as depicted in Figure 1a. As the carbothermal reduction temperature increased to 800 ℃, however, these particles started aggregating and bonding together, a phenomenon clearly visible in Figure 1b. Fur-
ther escalation of the carbothermal reduction temperature to 1500 ℃ led to the formation of a greater quantity of the RF/SiO2 ceramic particles with surfaces appearing notably smoother. This transformation could be attributed to the enhanced crystallinity of the
RF/SiO2 ceramic particles achieved at such an elevated temperature, as illustrated in Figure 1c.".
5) There are small grammatical mistakes in the work that need to be "combed" until they disappear. For example in the first part of section 3.1 "a phenomenon clearly visible in Figure 1b" should be written in the form "a phenomenon visible in Figure 1b". "Clearly" needs to be eliminated.
Or: write "To delve deeper into the phase transition dynamics, a crystal phase analysis was performed on samples RS-0, RS-800, and RS-1500, with their respective XRD patterns displayed in Figure 1d. In the case of RS-0, a broad diffraction peak within the range of 15 to 25° signified the presence of the amorphous SiO2 ceramic particles." as "To delve into phase transition dynamics, a crystal phase analysis was performed on samples RS-0, RS-800, and RS-1500, with their respective XRD patterns displayed in Figure 1d. In the case of RS-0, a broad diffraction peak within the range of 15 to 25° signified the presence of the amorphous SiO2 ceramic particles.".
Or: write "The FTIR spectrum of the RF/SiO2 ceramic particles was shown in Figure 1e. The peaks at 2887 cm-1, 1608 cm-1, and 1463 cm-1 were assigned to C-H stretching vibrations." as "The FTIR spectrum of the RF/SiO2 ceramic particles is shown in Figure 1e. The peaks at 2887 cm-1, 1608 cm-1, and 1463 cm-1 were assigned to C-H stretching vibrations.".
6) Explain the statement "The absence of any sharp, unique peaks in the XRD pattern of RS-800 following an increase in heat treatment temperature to 800 ℃ suggested that the ceramic particles retained their amorphous structure at this temperature level. However, upon subjecting the sample to a higher temperature of 1500 ℃, a prominent sharp peak emerged at 22.23°, which corresponded to the (101) crystal plane of SiO2. This observation conclusively indicated that a phase transformation had indeed occurred following the 1500 ℃ heat treatment." in more detail, or enter a justification in this sense or a citation.
7) Explain a little the statement: "The FTIR spectrum of the RF/SiO2 ceramic particles was shown in Figure 1e. The peaks at 2887 cm-1, 1608 cm-1, and 1463 cm-1 were assigned to C-H stretching vibrations." Details about the procedures used in this regard.
8) Some more explanations about the statement "Additionally, XPS analysis was employed to investigate the chemical structure
of the products obtained at different heat treatment temperatures, particularly quantitatively analyzing the structural changes of silicon elements at different heat treatment temperatures." Details about the procedures used in this regard.
9) Discuss in more detail the main elements of the first two figures 1-2.
10) In the end of section 3, in fact, the only developed section of the work, must discuss in more detail all the new aspects introduced in the work, with their advantages and possible disadvantages, if there are limitations, possible aspects that still need to be deepened, and possible future research opened by current methodology. Highlight the record aspects "electromagnetic absorption performance and high-temperature resistance".
11) Important: Explain in detail how you pursued "stability in performance over time" as this is essential for EMI mitigation, consistency, and reliability.
12) Develop the "Conclusion" section.
13) Enter multiple references.
Comments on the Quality of English Language
Minor editing of the English language is required
Reviewer 4 Report
Comments and Suggestions for Authors
The first of all the difference of submitted paper to showed published paper [a] must be clearly stated
[a] Xinli Ye, Hao Yu, Kai Zheng, Shan Li, Xiaomin Ma, Bangxiao Mao, Junxiong Zhang, Synthesis and microwave absorption performance of heat-treated RF/SiO2 aerogels, Defence Technology, Volume 34, 2024, Pages 177-186, ISSN 2214-9147, https://doi.org/10.1016/j.dt.2023.10.006.
These works looks too similar. For example fig 2 in {a} looks as a fragment of fig 1 in submitted work. The misspellings are about a legend, in one work we see the S800, in the other RS800, and they probably mean the same thing.
Moreover fig 2 in submitted work looks like ann other color version of figure 5 in [a].
From the above indicated (but not only these cases) it is difficult for me to refer to the originality of the submitted work. In my opinion, these papers are too close to each other for the paper submitted to MDPI to be accepted for publication. Otherwise, the work itself is interesting, but so is the work printed as [a].
Reviewer 5 Report
Comments and Suggestions for Authors
The document is attached

Comments on the Quality of English Language
Moderate editing of English language required. I do not like the use of the past in the text.
Round 2
Reviewer 1 Report
Comments and Suggestions for Authors
There are still no lines numbers for identification which again makes it very difficult for the review. This should have been addressed already and was ignored by the authors in the point-by-point rebuttal.
The authors need to check the journal abbreviations in their references as they are not consistent.
Figure s2 and 5 have labels SC-0, SC-8, and SC-15 which are not mentioned anywhere else in the manuscript and should be removed since the text refers to the samples as RS-0, RS-800, and RS-1500. This changing notation is confusing.
The provided hypothesis to the factors responsible for the observed excellent absorption properties of the RF/SiO2 particles (Response 4) needs to be included in the manuscript text not just in the rebuttal.
Why is reference 27 included in paragraph 4 on page 3 if this is an experimental result? Explain what is being cited from this reference.
The provided response to the lack of discussion of structural defect sites within absorbers (response 6) did not address including any discussion of structural defect sites in the manuscript text.
The provided response 11 does not include language in line 6 of the abstract in page 1 that the properties persist after the temperature action is eliminated.
Abstract
“successfully generated after heat treatment at 1500℃ After heat treatment at 1500 ℃, when” does not make sense.
Page 2
Should be “Subsequently, they”
The importance of the work in reference 23 is still not clearly explained. For example, were any applications permitted when “the electromagnetic wave absorption performance of the material [23]” was improved? What is the outcome of this improvement?
Page 3
low-cast should be low cost
The provided response 25 does not include addressing the human health risk factors associated with resorcinol formaldehyde in the manuscript and clarification that the substances are liquids
Page 4
The provided response 32 should change the manuscript language from bonding together to agglomeration in the paragraph text.
The provided response 34 should include the substrate used for XRD in the manuscript
The provided response 35 should include the standard diffraction data for alpha-SiO2 reference in the manuscript
“The FTIR spectrum of the RF/SiO2 ceramic particles is shown in Figure 1e.” should be “The FTIR spectra of the RF/SiO2 ceramic particles are shown in Figure 1e.”
The provided response 37 should include discussion regarding the FTIR spectra in the manuscript with regards to Si-O2 peak height if the authors can notated differences across the samples.
The provided response 38 should include modifications in the manuscript.
The provided response 39 did not add discussion in the manuscript to identify the blue (deconvoluted) spectra in Figs 1 f-h and discuss peak width.
The provided response 42 should include modifications in the manuscript on page 5 where discussed.
The provided response 43 should include an arrow added to figure 2b,e and the arrow referenced in the manuscript when discussed on the bottom of page 5, if too difficult for the reader to discern but noted as an important in the performance section of the paper.
Response 44 Thickness and frequency are still too difficult to read in Fig. 2a-c. Something is wrong with the font, it is distorted.
Page 7
The provided response 48 should include modifications in the text to identify why dashed lines are included in Figure 3
Re-word as this sentence does not make sense: “It is generally believed that the larger attenuation constant (α) value is, the greater ability of the absorber attenuates the EMWs [35].”
The provided response 50 should include modifications in the text regarding the 1.0 impendence matching that was also reached at frequencies of 5 and 7 GHz.
Comments on the Quality of English Language
Some minor edits needed as pointed out above
Reviewer 2 Report
Comments and Suggestions for Authors
The authors have corrected and improved the manuscript according to my suggestions. The manuscript can be accepted in this form.
Author Response
Thank you for acknowledging our article and your recognition not only serves as a significant encouragement but also validates our efforts and work. We will strive to continue producing valuable content.
Reviewer 3 Report
Comments and Suggestions for Authors
Accept in present form
Comments on the Quality of English Language
Minor editing of the English language is required
Author Response
Thank you for your acknowledgment of our article; we have already devoted considerable effort to refining its language and making meticulous revisions. Your positive feedback is invaluable and motivates us to maintain our commitment to excellence in content creation.
Reviewer 4 Report
Comments and Suggestions for Authors
Please include such or similar an explanation
“Response 1: ….., these two works” (actual and Xinli Ye, Hao Yu, Kai Zheng, Shan Li, Xiaomin Ma, Bangxiao Mao, Junxiong Zhang, Synthesis and microwave absorption performance of heat-treated RF/SiO2 aerogels, Defence Technology, Volume 34, 2024, Pages 177-186, ISSN 2214-9147, https://doi.org/10.1016/j.dt.2023.10.006) “ were carried out concurrently, with the raw materials and processes being generally similar, but there are certain distinctions: Firstly, the heat treatment atmospheres are different, with the former conducted under a 90% nitrogen atmosphere and the latter processed in pure nitrogen. According to the material system of the former, sintering at 1500°C in a pure nitrogen atmosphere yields SiC material. However, in this work, we added a small amount of aluminum isopropoxide as an additive and, surprisingly, found that after heat treatment at 1500°C, no SiC was formed in comparison to when the additive was not included. We suspect that the addition of aluminum isopropoxide may have hindered the formation of SiC. The underlying mechanism behind this phenomenon is currently under our gradual investigation.”
”
at the end of the introduction section. And then clearly state the purpose of the work, as you put in your response to the previous review.
I think that such a description of the diversity of the results of similar works is necessary at the outset. Positioning it in the discussion section area is too late for the reader.
After such an explanation, the remaining corrections made so far indicate the independence of both works.
And these are interesting works, and their results are worthy of publication.
Author Response
Thank you for recognizing our article. We have come to realize the importance of stating the purpose and distinctions of our work from the outset. Your feedback is instrumental in helping us improve, and we appreciate it greatly. Once again, thank you for your attention and input—it is truly invaluable.
Reviewer 5 Report
Comments and Suggestions for Authors
The authors worked on enhancing the document and have provided comprehensive responses to all requests.
The work is ready for publication.
I suggest only removing the bullet points from the conclusion and instead crafting it in a more narrative style.
Author Response
Thank you for your appreciation of our article. The key points in the conclusion were indeed revised based on feedback from other reviewers. We hope these adjustments meet your satisfaction. However, if you feel further modifications are necessary, please do not hesitate to share your specific concerns or requirements. We are more than willing to make additional changes to align with your expectations. Your guidance is crucial in ensuring the quality and accuracy of our work.